# Floquet Engineering of Non-Equilibrium Superradiance

L. Broers[1,2,*] and L. Mathey[1,2,3]

**1** Center for Optical Quantum Technologies, University of Hamburg, Hamburg, Germany
**2** Institute for Laser Physics, University of Hamburg, Hamburg, Germany
**3** The Hamburg Center for Ultrafast Imaging, Hamburg, Germany
**\*** lbroers@physnet.uni-hamburg.de

March 16, 2022

## Abstract

We demonstrate the emergence of a non-equilibrium superradiant phase in the dissipative Rabi-Dicke model. This phase is characterized by a photonic steady state that oscillates with a frequency close to the cavity frequency, in contrast to the constant photonic steady state of the equilibrium superradiant phase in the Dicke model. We relate this superradiant phase to the population inversion of Floquet states by introducing a Schwinger representation of the driven two-level systems in the cavity. This inversion is depleted near Floquet energies that are resonant with the cavity frequency to sustain a coherent light-field. In particular, our model applies to solids within a two-band approximation, in which the electrons act as Schwinger fermions. We propose to use this Floquet-assisted superradiant phase to obtain controllable optical gain for a laser-like operation.

# 1   Introduction

Driven dissipative quantum systems display a plethora of intriguing phenomena, including unconventional coherent light sources and amplification mechanisms. Phenomena such as lasing without inversion [1–4], lasing with driven quantum dots [5,6] and population inversion in strongly driven two-level systems [7], have been proposed or implemented to extend conventional lasing. These examples are based on the non-equilibrium dynamics of the dissipative Rabi model, which presents a minimal example of driven quantum systems. Similarly, driven Dicke models [8] exhibit rich non-equilibrium dynamics of superradiant phase transitions and unconventional lasing states [9–23]. In many-body systems, Floquet engineering aims to tune collective properties, such as band topology [24–28], with coherent driving [29–31]. It has been shown that population inversion of Floquet states can occur in driven systems [32–35]. Floquet theory itself presents a method to describe the effective dressed states in driven systems and their population, and is in particular also applicable to driven dissipative cavity systems [36, 37].

We present the emergence of a Floquet-assisted superradiant phase (FSP) in the dissipative Dicke model under the influence of circularly polarized driving of the two-level systems, reminiscent of the Rabi model. This superradiant phase presents a mechanism for light-amplification and coherent light sources in two-level systems that is induced by the driven coherences between effective dressed states and is thus not captured by semi-classical rate equations in which population inversion is impossible. We find that this mechanism originates from the effective population inversion of Floquet states that is depleted and transferred into the cavity if the cavity frequency is close to resonance with the Floquet energy difference. This photonic coherent state saturates quickly, leading to a steady state of constant magnitude with respect to the coupling strength. We analytically determine the regime of driving field strengths in which the system displays Floquet state population inversion and is therefore susceptible to the FSP. We further predict the onset of the FSP in the limit of small coupling strengths.

This work demonstrates that despite the fact that Floquet states are effective descriptions with energies that are only defined modulo multiples of a given driving frequency, their population inversion can induce and sustain a coherent photonic state in a close-to-resonant cavity. The connection between this light amplification mechanism in two-level systems and effective populations of Floquet states translates into solid state systems that can be described with two bands, e.g. monolayer graphene. This suggests the possibility of coherent Floquet engineered light-amplification in solids.

This work is structured as follows. In section 2, we describe the Rabi-Dicke model and its dissipative mean-field description. In section 3, we present numerical results for the phase diagram of the photonic steady state which shows the FSP. We also show the photonic steady state of the FSP in frequency space as a function of the driving field strength. Further, we present analytical calculations of the Dicke superradiant transition in this model. In section 4, we extend our results to a Schwinger representation which we use to calculate two-point correlation functions and Floquet state populations. In this representation we see the population inversion of the Floquet states and its depletion in the FSP. We then present an approximation of the Floquet energies of the two-level system in the FSP from an approximate bichromatic Floquet description. In section 5, we present analytical bounds for the driving field strengths at which population inversion occurs. Additionally, we demonstrate

an accurate description of the onset at which the FSP first occurs for weak coupling to the cavity. In section 6, we conclude and discuss our findings.

## 2   Dissipative Rabi-Dicke Model

We consider a system of $N$ identical two-level systems with level-spacing $\omega_z$ coupled to a single lossy cavity mode with frequency $\omega_c$, as schematically depicted in Fig. 1. We emphasize that the dynamical superradiant state can be realized on any set of well-defined two-level systems, including solids in a two-band approximation, see e.g. [38]. The individual two-level systems experience Rabi-like driving with frequency $\omega_d$ and field strength $E_d$. The Hamiltonian of this Rabi-Dicke model is

$$H = \sum_{j=1}^{N}[\frac{\omega_z}{2}\sigma_z^j + \frac{E_d}{\omega_d}(\sigma_+^j e^{-i\omega_d t} + \sigma_-^j e^{i\omega_d t})] + \omega_c a^\dagger a + \frac{2\lambda}{\sqrt{N}}\sum_{j=1}^{N}(a + a^\dagger)\sigma_x^j, \tag{1}$$

where $\lambda$ is the coupling strength and $\sigma_{x,y,z}^j$ are the Pauli-matrices of the $j$th two-level system. It is $\sigma_\pm = (\sigma_x \pm i\sigma_y)/2$. $a^{(\dagger)}$ is the photon annihilation (creation) operator.

We use a mean-field approximation of the photon dynamics via the coherent state ansatz $\alpha = \alpha_r + i\alpha_i = \langle a \rangle$, with the system separating into the two-level subsystem A and the cavity subsystem C resulting in the approximate Hamiltonian $H = \sum_j H_A^j + H_C$, with

$$H_A^j = \frac{\omega_z}{2}\sigma_z^j + \frac{E_d}{\omega_d}(e^{-i\omega_d t}\sigma_+^j + e^{i\omega_d t}\sigma_-^j) + \frac{2\lambda\langle a + a^\dagger \rangle}{\sqrt{N}}\sigma_x^j \tag{2}$$

$$H_C = \omega_c a^\dagger a + 2\lambda\sqrt{N}\langle\sigma_x\rangle(a + a^\dagger). \tag{3}$$

We include a cavity loss rate $\kappa$, such that the equation of motion of the photon mode is

$$\dot{\alpha} = -(i\omega_c + \kappa)\alpha - i2\lambda\sqrt{N}\langle\sigma_x\rangle. \tag{4}$$

The Lindblad-von Neumann master equation of the two-level system is

$$\dot{\rho} = i[\rho, \frac{\omega_z}{2}\sigma_z + \frac{E_d}{\omega_d}(e^{-i\omega_d t}\sigma_+ + e^{i\omega_d t}\sigma_-) + \frac{2\lambda\alpha_r}{\sqrt{N}}\sigma_x] + \sum_{l\in\{+,-,z\}}\gamma_l[L_l\rho L_l^\dagger - \frac{1}{2}\{L_l^\dagger L_l, \rho\}], \tag{5}$$

where we omit the superscript $j$, since the two-level systems are all identical, in this approximation. We describe the dissipation of the two-level system in its instantaneous eigenbasis. In particular, the Lindblad operators are $L_+ = V\sigma_+V^\dagger$, $L_- = V\sigma_-V^\dagger$ and $L_z = V\sigma_zV^\dagger$, where $V$ is the unitary transformation into the instantaneous eigenbasis of $H_A(t) = \epsilon_A(t)V\sigma_zV^\dagger$. $\epsilon_A(t)$ is the instantaneous eigenenergy of the Hamiltonian $H_A(t)$. $\gamma_\pm$ and $\gamma_z$ are the coefficients of spontaneous decay and dephasing, respectively. The equation of motion of the two-level system then takes the form (see App. A)

$$\dot{\rho} = i[\rho, H_A(t)] - \gamma_1\rho - \gamma_2 H_A(t)\epsilon_A^{-1}(t) - \frac{1}{2}\gamma_3\text{Tr}(\rho H_A(t))H_A(t)\epsilon_A^{-2}(t) \tag{6}$$

with

$$\gamma_1 = (\gamma_- + \gamma_+ + \gamma_z)/2 \qquad \gamma_2 = (\gamma_- - \gamma_+)/2 \qquad \gamma_3 = (\gamma_- + \gamma_+ - \gamma_z)/2. \tag{7}$$

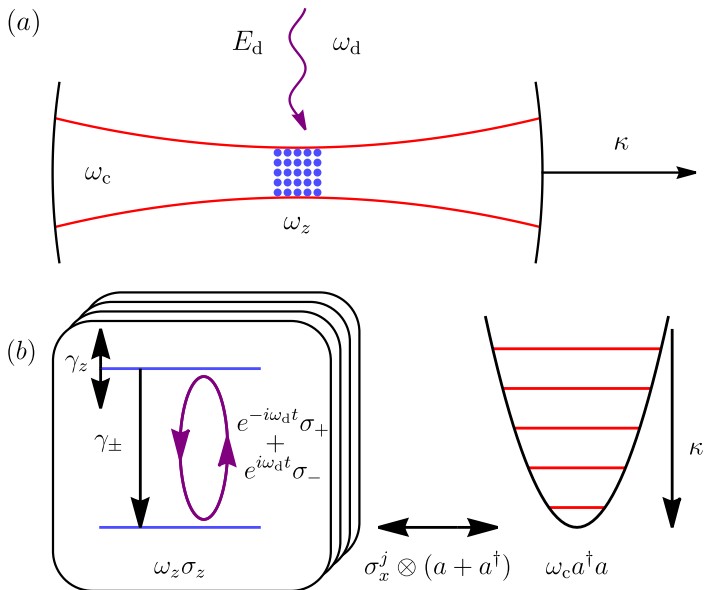

Figure 1: An illustration of the dissipative Rabi-Dicke model (a) and a depiction of its Hamiltonian as in Eq. 1 (b). A cavity (red) contains a set of identical two-level systems (blue) which experience circularly polarized Rabi-like driving (purple). $\gamma_\pm$ and $\gamma_z$ denote the coefficients of dissipative processes in the two-level systems, i.e. spontaneous decay and dephasing. $\kappa$ is the loss rate of the cavity, which determines the coherent output of the cavity.

Throughout this work we use the values $\gamma_- \approx \frac{\omega_d}{100\pi}$, $\gamma_+ \approx 0$, $\gamma_z \approx \frac{\omega_d}{45\pi}$ and $\kappa \approx \frac{\omega_c}{100}$. The two-level losses are small and correspond to the regime in which Floquet states form. The cavity loss rate is small compared to what is commonly referred to as the 'good cavity' regime. We find that our results are sensitive to these dissipation coefficients. However, the scaling behavior with respect to dissipation is not the focus of this work. This choice of the dissipative model is motivated by the natural dissipative environment of electrons in a solid, see Ref. [38]. The two-level systems that we consider here can be realized as two electron states, with one electron occupying one or the other. As we describe below, these two states can be embedded in a four-level system that includes both states to be occupied or empty, within a Schwinger construction. While this is the natural Hilbert space for an electronic realization, we emphasize that the results we obtain here can be generated from the Rabi-Dicke model, i.e. Eq. 1.

## 3    Floquet-Assisted Superradiant Phase

We determine the steady state regimes of the system. For that purpose, we solve the equations of motion Eqs. 4 and 5 and find the photonic state $\alpha(t)$, which serves as the order parameter of superradiant phases. In Fig. 2 (a), we show the magnitude of $\alpha$ as a function of the driving field strength $E_d$ and the coupling strength $\lambda$, for $\omega_z = \omega_d/2$ and $\omega_c = \omega_d/4$, as an example. We note that no specific ratio between these frequencies is required. We find two phases of non-zero $|\alpha|$. The phase for small driving field strengths $E_d$ is related to the Dicke

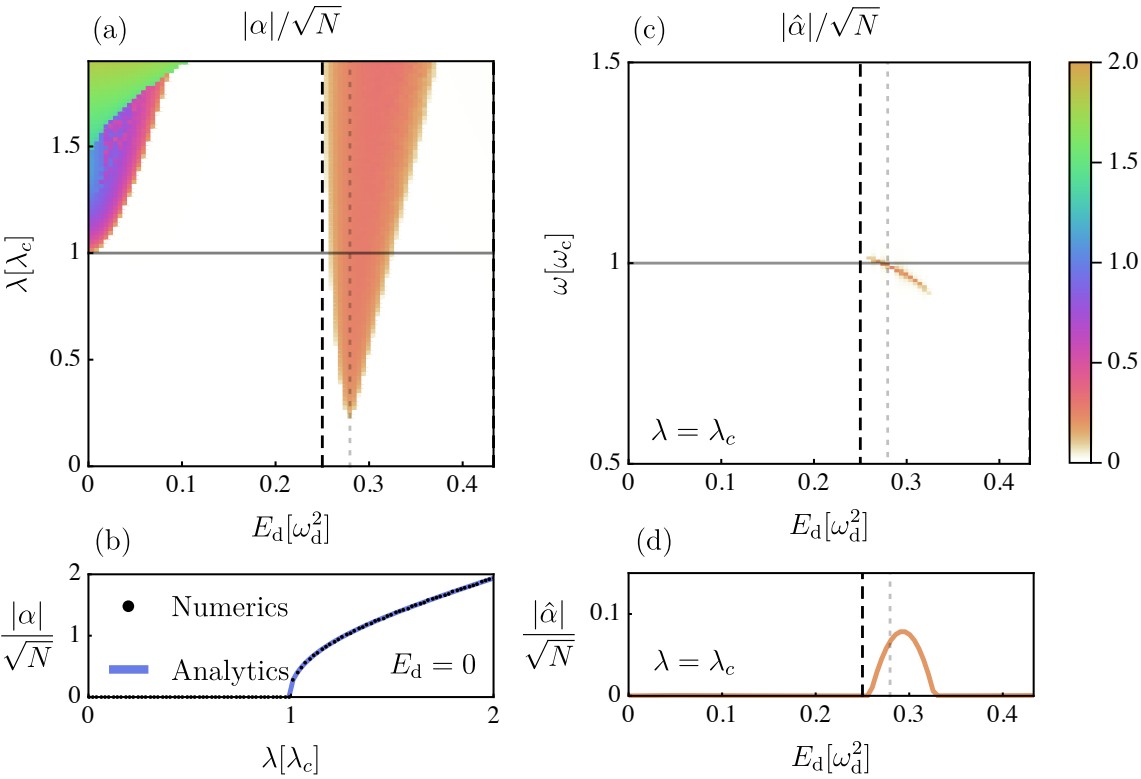

Figure 2: In panel (a) we show the magnitude of the photonic field $\alpha$ as a function of the driving field strength $E_\mathrm{d}$ and the coupling strength $\lambda$. For large $E_\mathrm{d}$, the Floquet-assisted superradiant phase (FSP) emerges and exhibits an oscillating photonic steady state. In panel (b) we show the $E_\mathrm{d} \to 0$ limit, i.e. the Dicke superradiant transition, which is predicted very well analytically. In panel (c) we show the magnitude of the Fourier transform $|\hat{\alpha}|/\sqrt{N}$ as a function of the driving field strength $E_\mathrm{d}$ for the coupling strength $\lambda = \lambda_c$. In the FSP, the steady state frequency of the cavity is close to the cavity frequency. In panel (d) we show the integrated power spectrum of the FSP as a function of the driving field strength $E_\mathrm{d}$. The dashed lines in (a), (c) and (d) indicate the analytically determined lower bound for the FSP, see Eq. 23. The dotted lines in (a), (c) and (d) indicate the driving field strength at which the Floquet energy spacing is equal to the cavity frequency.

superradiant phase and approaches it for $E_{\mathrm{d}} \to 0$, which is an equilibrium phenomenon. In this limit, Eq. 1 recovers the dissipative Dicke-model. To capture this state, we write the equilibrium state of the static two-level system as

$$\rho = \frac{1}{2}(\mathbb{1} - \frac{\gamma_- - \gamma_+}{\gamma_- + \gamma_+}\frac{H_{\mathrm{A}}}{\epsilon_{\mathrm{A}}}), \tag{8}$$

which solves Eq. 6. We find the corresponding photonic steady state from Eq. 4 by inserting $\dot{\alpha} = 0$ and $\langle \sigma_x \rangle = \rho_x$. It is

$$0 = -(i\omega_{\mathrm{c}} + \kappa)(\alpha_r + i\alpha_i) - i2\lambda\sqrt{N}\rho_x \tag{9}$$

with

$$\rho_x = -\frac{\gamma_- - \gamma_+}{\gamma_- + \gamma_+}\frac{4\lambda\alpha_r N^{-\frac{1}{2}}}{\sqrt{\omega_z^2 + 16\lambda^2\alpha_r^2 N^{-1}}}, \tag{10}$$

which we solve to find

$$\frac{\alpha}{\sqrt{N}} = (1 + i\frac{\kappa}{\omega_{\mathrm{c}}})\sqrt{\left(\frac{\gamma_- - \gamma_+}{\gamma_- + \gamma_+}\frac{2\lambda\omega_{\mathrm{c}}}{\omega_{\mathrm{c}}^2 + \kappa^2}\right)^2 - \left(\frac{\omega_z^2}{4\lambda^2}\right)^2}. \tag{11}$$

If $\alpha$ is purely imaginary, then $\rho_x$ is zero, because of Eq. 10. This implies that the $\alpha = 0$ solution is the state of the system, based on Eq. 9. If $\alpha$ has a non-vanishing real part, i.e. $\alpha_r \neq 0$, the system is in the Dicke superradiant state. We determine the critical coupling strength $\lambda_c$ of this transition by setting the expression under the root in Eq. 11 equal to zero. It is

$$\lambda_c = \frac{1}{2\sqrt{2}}\sqrt{\frac{\gamma_- + \gamma_+}{\gamma_- - \gamma_+}\frac{\omega_z}{\omega_{\mathrm{c}}}\left(\kappa^2 + \omega_{\mathrm{c}}^2\right)}. \tag{12}$$

In the case of $\kappa = 0$ and $\gamma_+ = \gamma_- e^{-\frac{\omega_z}{2k_B T}}$ this reproduces the well-known result for the critical coupling

$$\lambda_c = \frac{1}{2\sqrt{2}}\sqrt{\omega_z\omega_{\mathrm{c}}\coth(\frac{\omega_z}{4k_B T})} \overset{T \to 0}{\to} \frac{\sqrt{\omega_z\omega_{\mathrm{c}}}}{2\sqrt{2}}. \tag{13}$$

We show this transition in Fig. 2 (b) compared to the numerical solution, which show excellent agreement. Increasing $E_{\mathrm{d}}$ initially maintains this transition, but increases the critical coupling strength $\lambda_c|_{E_{\mathrm{d}}>0} - \lambda_c \propto E_{\mathrm{d}}^2$. Further, the phase is separated into two regimes by a boundary $\lambda_{\mathrm{b}}$. For $E_{\mathrm{d}} > 0$ and $\lambda < \lambda_{\mathrm{b}}$ the phase shows similar scaling beyond the transition as for $E_{\mathrm{d}} = 0$. However, the system also experiences heating in this part of the phase. For $\lambda > \lambda_{\mathrm{b}}$ the superradiant phase is constant with respect to $E_{\mathrm{d}}$, i.e. the value of $\alpha$ matches the case of $E_{\mathrm{d}} = 0$. In Fig. 2 (a), we show that this separation occurs for $E_{\mathrm{d}} = 0$ at $\lambda_{\mathrm{b}} \approx 1.5\lambda_c$ for the given parameters and increases with increasing $E_{\mathrm{d}}$.

For larger field strengths $E_{\mathrm{d}}$, there is a second superradiant phase, the FSP, with a non-zero photon amplitude $|\alpha|$. The existence and properties of this non-equilibrium state is the central point of this paper. For weak coupling, i.e. $\lambda \ll \lambda_c$, this phase emerges at the driving field strength at which the difference of Floquet quasi-energies is resonant with the cavity mode, as we discuss later. For increasing $\lambda$, this domain broadens and gives the tongue structure in Fig. 2 (a). Within this phase, $|\alpha|$ quickly approaches a constant value for increasing coupling strength $\lambda$. The dashed line in Fig. 2 (a) indicates the asymptotic lower bound of the FSP for increasing $\lambda$. We calculate and present the driving field strengths that bound the FSP in section 5.

In Fig. 2 (c) we show the magnitude of the Fourier transform $\hat{\alpha}(\omega)$ of the photonic steady state as a function of the driving field strength $E_d$ at $\lambda = \lambda_c$, indicated by the solid line in Fig. 2 (a). We see that the steady state of the cavity in the FSP oscillates with a frequency close to the cavity frequency $\omega_c$, as indicated by the solid horizontal line. This differs from the Dicke superradiant phase in which the steady state is not oscillatory. The frequency in the FSP is the effective Floquet energy difference of the two-level system, which is interacting non-linearly with the cavity mode, as we elaborate in the following section. This energy is equal to the cavity frequency $\omega_c$ at the driving field strength indicated by the vertical dotted lines, which is the same as the onset driving field strength at which the FSP emerges for small $\lambda$ in Fig. 2 (a). In Fig. 2 (d) we show the total intensity of the photon mode, which corresponds to the integrated power spectrum $\int |\hat{\alpha}(\omega)|^2 N^{-1} d\omega$ of the Fourier transform of $\alpha$. In the following section, we show that this profile of the magnitude of the order parameter is related to the depleted population inversion of the Floquet states of the two-level system.

## 4 Floquet State Population Inversion

To understand the underlying mechanism from which the FSP originates, we calculate the Floquet state population of the driven two-level system. We introduce a Schwinger representation of the two-level Hamiltonian in Eq. 2, and calculate the population in frequency space. In this representation the system is embedded into a larger system consisting of two modes $b_1$ and $b_2$. The resulting Hilbert-space is spanned by the creation operators $b_1^\dagger$ and $b_2^\dagger$ of these two modes. Note that these modes can be understood as hard-core bosons in the atomic case of the Dicke model, i.e. $b_1^2 = b_2^2 = 0$, but also as fermions in two-band models of solid state systems, where these are the electrons, cp. [35,38]. Our mean-field results are not affected by the specific exchange relations, bosonic or fermionic. The Pauli-matrices are written as

$$\sigma_x = b_1^\dagger b_2 + b_2^\dagger b_1 \qquad\qquad \sigma_y = i(b_1^\dagger b_2 - b_2^\dagger b_1) \qquad\qquad \sigma_z = b_1^\dagger b_1 - b_2^\dagger b_2. \qquad (14)$$

We calculate the two-point correlation functions $\langle b_j^\dagger(t_2) b_j(t_1)\rangle$ and determine the frequency resolved population of the two-level steady state as

$$n(\omega) = \frac{1}{(\tau_2 - \tau_1)^2} \int_{\tau_1}^{\tau_2} \int_{\tau_1}^{\tau_2} \sum_{j=1}^{2} \langle b_j^\dagger(t_2) b_j(t_1)\rangle \, e^{-i\omega(t_2 - t_1)} dt_2 dt_1, \qquad (15)$$

where the time $\tau_1$ is large enough for the system to have reached a steady state and $(\tau_2 - \tau_1)$ is large enough to contain hundreds of driving periods. Note that in this calculation the operators $b_j(t_1)$ and $b_j^\dagger(t_2)$ act only on one of the $N$ atoms. For large $N$, we assume that the remaining $N-1$ atoms maintain their steady state unaltered, such that the steady state $\alpha(t)$ is also not affected by either action of $b_j(t_1)$ or $b_j^\dagger(t_2)$.

We show $n(\omega)$ as a function of the driving field strength $E_d$ in Fig. 3 (a) for $\lambda = \lambda_c$. We use the same values of $\omega_z = \omega_d/2$ and $\omega_c = \omega_d/4$ as for the example in Fig. 2. We see that the state of the probed two-level system is distributed across frequencies that are resonant with the Floquet energies of the system and its replicas $\pm\epsilon_F^0 + m\omega_d$, $m \in \mathbb{Z}$. For $\lambda = 0$, and $\alpha = 0$, these Floquet energies are

$$\epsilon_F^0 = \frac{\omega_d}{2} \pm \sqrt{\frac{E_d^2}{\omega_d^2} + \frac{(\omega_d - \omega_z)^2}{4}}. \qquad (16)$$

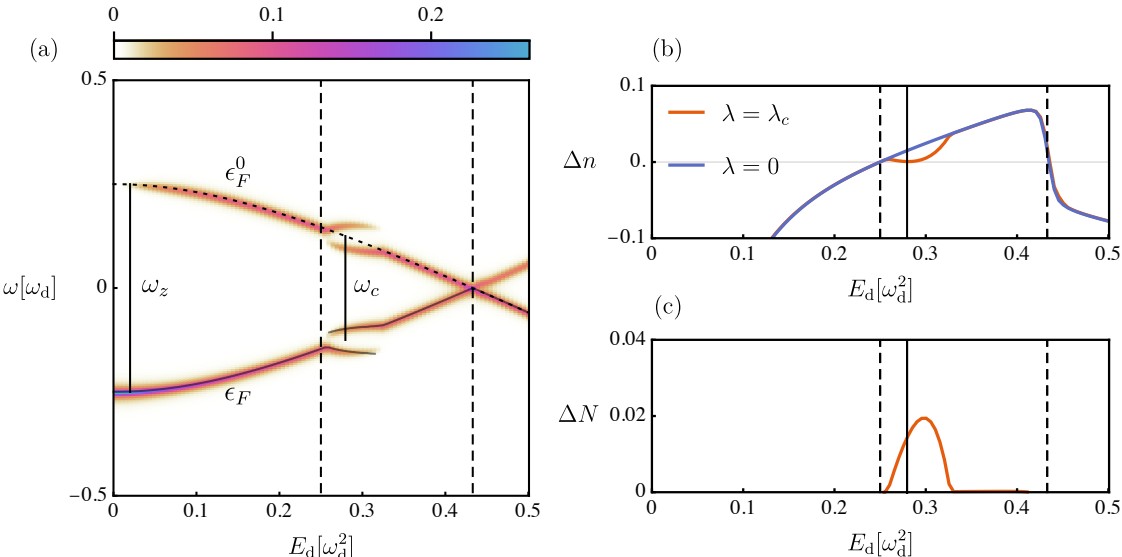

Figure 3: In panel (a) we show the Floquet state population $n(\omega)$ as a function of the driving field strength $E_{\mathrm{d}}$ calculated in the Schwinger formalism. The dotted line indicates the Floquet energies $\epsilon_F^0$ for $\lambda = 0$, the solid gray lines indicate the approximate Floquet energies $\epsilon_F$ for $\lambda = \lambda_c$ which results from Eq. 19. In panel (b) we show the effective population difference $\Delta n$ between Floquet states for $\lambda = 0$ (blue) and $\lambda = \lambda_c$ (red). The regime in which population inversion occurs also contains the FSP, which depletes the inversion. In panel (c) we show the difference $\Delta N$ between the two populations in panel (b). The dashed lines in all panels indicate the values of $E_{\mathrm{d}}$ that bound the regime in which population inversion occurs. The solid lines in (b) and (c) indicate the driving field strength at which the Floquet energy difference $\Delta \epsilon_F^0$ is resonant with the cavity frequency $\omega_c$.

In the regime of the FSP, the Floquet spectrum is modified due to the additional driving that the two-level system experiences from the interaction with the oscillating photonic steady state. We approximate that the FSP oscillates at $\omega_c = \omega_d/4$. The integer ratio of $\omega_d$ and $\omega_c$ is not required, it merely enables a two-frequency Floquet analysis. For this choice of frequencies the two-level Hamiltonian in Eq. 2 is

$$H(t) = e^{-i4\omega_c t}H_{-4} + e^{-i\omega_c t}H_{-1} + H_0 + e^{i\omega_c t}H_1 + e^{i4\omega_c t}H_4 \tag{17}$$

with

$$H_0 = \frac{\omega_z}{2}\sigma_z \qquad H_{\pm 1} = \frac{\lambda|\alpha|}{\sqrt{N}}\sigma_x \qquad H_{\pm 4} = \frac{E_d}{\omega_d}\sigma_{\mp}. \tag{18}$$

The corresponding Floquet Hamiltonian is

$$H_F = \begin{pmatrix} \ddots & H_1 & & & & H_4 & & \\ H_{-1} & H_0 + 2\omega_c & H_1 & & & & H_4 & \\ & H_{-1} & H_0 + \omega_c & H_1 & & & & H_4 \\ & & H_{-1} & H_0 & H_1 & & & \\ H_{-4} & & & H_{-1} & H_0 - \omega_c & H_1 & & \\ & H_{-4} & & & H_{-1} & H_0 - 2\omega_c & H_1 & \\ & & H_{-4} & & & & H_{-1} & \ddots \end{pmatrix}. \tag{19}$$

It operates on the Floquet representation of the state

$$|\psi\rangle\rangle = (\ldots, \psi_{\uparrow,(n-1)\omega_c}, \psi_{\downarrow,(n-1)\omega_c}, \psi_{\uparrow,n\omega_c}, \psi_{\downarrow,n\omega_c}, \ldots)^{\mathrm{T}}. \tag{20}$$

Inserting the solutions of $\alpha$ that we show in Fig. 2 (a), allows us to calculate the Floquet energies $\epsilon_F$ in the FSP. We show these Floquet energies as a function of the driving field strength $E_d$ in Fig. 3 (a) as gray solid lines. We see that these energies match the dominantly populated frequencies in $n(\omega)$ of the two-level system very well. Note that slight mismatches are a consequence of the approximation made to justify the expression of $H_F$.

We sum up the population of all Floquet replicas to calculate the effective relative population of the two-level system as

$$\Delta n = \sum_{m=-\infty}^{\infty} \left[ \int_{m\omega_d}^{(m+\frac{1}{2})\omega_d} n(\omega)\mathrm{d}\omega - \int_{(m+\frac{1}{2})\omega_d}^{(m+1)\omega_d} n(\omega)\mathrm{d}\omega \right]. \tag{21}$$

In Fig. 3 (b), we show this effective relative population $\Delta n$ of the two-level system as a function of the driving field strength $E_d$ for the cases of $\lambda = 0$ and $\lambda = \lambda_c$. We see that there is a regime in which the system experiences an effective population inversion, bracketed by the vertical dashed lines. In the case of non-zero coupling, i.e. $\lambda = \lambda_c$, part of the population inversion is partially depleted to maintain the FSP, i.e. the non-zero steady state of the photon mode. In Fig. 2 (a), we see that the range of the FSP increases for increasing values of $\lambda$, to approach the entire regime in which population inversion occurs. In general, the FSP regime is smaller than the inversion regime, because of the detuning of the cavity frequency $\omega_c$ and the Floquet quasi-energy difference $\Delta\epsilon_F^0$.

In Fig. 3 (c), we show the depletion of the effective population inversion of the two-level system

$$\Delta N = \Delta n|_{\lambda=0} - \Delta n|_{\lambda=\lambda_c}. \tag{22}$$

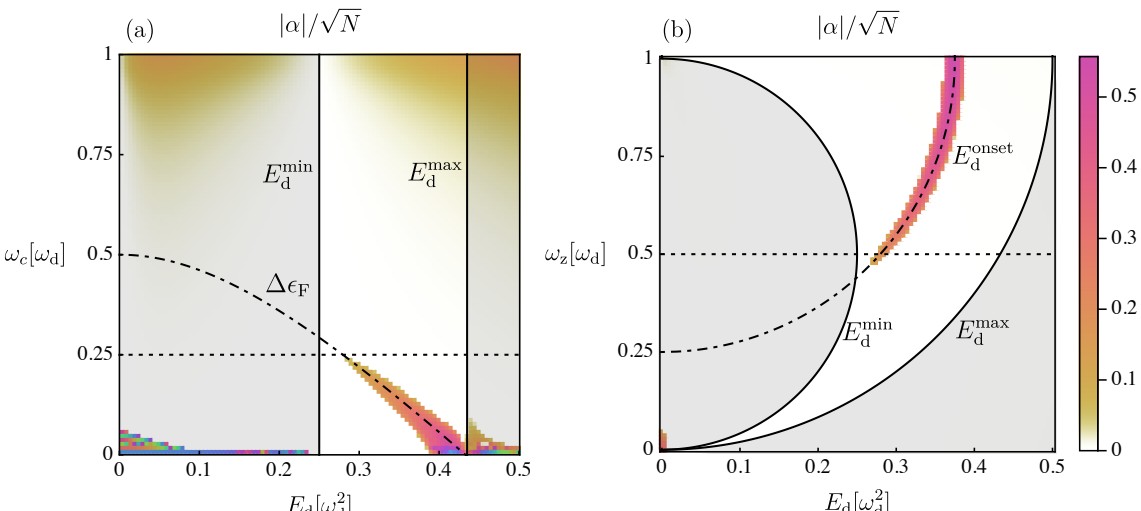

Figure 4: The magnitude of the photonic steady state $\alpha$ as a function of the driving field strength $E_\mathrm{d}$, the cavity frequency $\omega_\mathrm{c}$ (a) and the two-level energy spacing $\omega_z$ (b). The coupling is small with $\lambda = \lambda_c/24$, such that the Floquet-assisted superradiant phase (FSP) appears only close to resonance between the cavity frequency $\omega_\mathrm{c}$ and the Floquet energy difference $\Delta\epsilon_F^0$, indicated by dot-dashed lines at $E_\mathrm{d}^\mathrm{onset}$. The gray shaded areas are regimes in which no population inversion of Floquet states occurs. They are bounded by $E_\mathrm{d}^\mathrm{min}$ and $E_\mathrm{d}^\mathrm{max}$. The dotted lines indicate the values of $\omega_\mathrm{c}$ and $\omega_z$ of the other subfigure, respectively.

The behavior of $\Delta N$ agrees very well with that of the photonic steady state that we show in Fig. 2 (d) up to an overall scale. We conclude that the photonic steady state of the FSP originates from the effective population inversion of the Floquet states which is depleted to obtain a non-zero $\alpha$. This explains the constant scaling of the FSP with respect to $\lambda$. In the limit of $\lambda \to \infty$, the intensity of the photonic steady state is limited by the population inversion of the Floquet states. Additionally, for large $\lambda$ the Dicke superradiant phase extends to larger values of $E_\mathrm{d}$, see Fig. 2 (a), which competes with the FSP.

## 5   Cavity-Resonant Floquet Energies

While the magnitude of the photon amplitude $\alpha$ saturates quickly to a constant value with increasing $\lambda$, here we determine the onset of the FSP for small $\lambda$. For small $\lambda$, the FSP emerges near resonance of the Floquet energy difference $\Delta\epsilon_F^0$ and the cavity frequency $\omega_\mathrm{c}$. We therefore present the dependence of the magnitude of $\alpha$ on the cavity frequency $\omega_\mathrm{c}$, as well as the two-level energy spacing $\omega_z$. In Fig. 4 (a) we show the magnitude of $\alpha$ as a function of the driving field strength $E_\mathrm{d}$ and the cavity frequency $\omega_\mathrm{c}$ at $\omega_z = \omega_\mathrm{d}/2$ and $\lambda = \lambda_c/24$. We see that the FSP emerges near resonance of $\Delta\epsilon_F^0$ and $\omega_\mathrm{c}$ with the lower bound of $E_\mathrm{d}$ given by the regime of the population inversion of Floquet states. For $\omega_\mathrm{c} \to 0$, the critical coupling $\lambda_c$

decreases to values smaller than that of $\lambda$ used here, such that we see the Dicke superradiant phase for small $E_\mathrm{d}$. For $\omega_\mathrm{c} \to \omega_\mathrm{d}$ we see an expected finite population in the cavity as it becomes resonant with the driving field.

We find the analytical solutions of the driven dissipative steady state for $\lambda = 0$ (See App. A) and use them to calculate the driving field strength at which the population inversion occurs ($E_\mathrm{d}^\mathrm{min}$). We also calculate the driving field strengths at which the Floquet state energies cross ($E_\mathrm{d}^\mathrm{max}$) and at which the Floquet energy difference is resonant with the cavity frequency ($E_\mathrm{d}^\mathrm{onset}$). They are

$$E_\mathrm{d}^\mathrm{min} = \omega_\mathrm{d}^2 \sqrt{\frac{1}{4} - \left(\frac{\gamma_1}{\omega_\mathrm{d}}\right)^2 + \left(\frac{1}{2} - \frac{\omega_z}{\omega_\mathrm{d}}\right)^2} \tag{23}$$

$$E_\mathrm{d}^\mathrm{max} = \omega_\mathrm{d}^2 \sqrt{1 - \left(1 - \frac{\omega_z}{\omega_\mathrm{d}}\right)^2} \tag{24}$$

$$E_\mathrm{d}^\mathrm{onset} = \omega_\mathrm{d}^2 \sqrt{\left(1 - \frac{\omega_\mathrm{c}}{\omega_\mathrm{d}}\right)^2 - \left(1 - \frac{\omega_z}{\omega_\mathrm{d}}\right)^2}. \tag{25}$$

Only the regime bound by $E_\mathrm{d}^\mathrm{min}$ and $E_\mathrm{d}^\mathrm{max}$ exhibits Floquet state population inversion in the two-level system and is therefore susceptible to the FSP for large enough $\lambda$. $E_\mathrm{d}^\mathrm{onset}$ indicates where the FSP first emerges for small $\lambda$, i.e. the driving field strength at which the Floquet energy difference is resonant with the cavity frequency. In Fig. 4 (b) we show these regimes and the magnitude of $\alpha$ as a function of the driving field strength $E_\mathrm{d}$ and the two-level spacing $\omega_z$ at $\omega_\mathrm{c} = \omega_\mathrm{d}/4$ and $\lambda = \lambda_c/24$. We see that $E_\mathrm{d}^\mathrm{onset}$ correctly predicts the initial onset of the FSP for small $\lambda$ inside the region of Floquet state population inversion.

# 6  Conclusion

We have demonstrated the emergence of a Floquet-assisted superradiant phase (FSP) in the dissipative Rabi-Dicke model that is directly related to the effective Floquet state population inversion of the two-level system. We propose to tune the Floquet energy difference close to resonance with the cavity, which results in the emergence of the FSP. In the FSP, the population inversion is depleted to populate a coherent photonic steady state that oscillates with a frequency that is close to the cavity frequency. This frequency is the Floquet energy difference of the effectively bichromatically driven two-level systems.

We have presented the frequency resolved state population of the two-level system, calculated in a Schwinger representation, and found that the depletion of the population inversion qualitatively agrees with the magnitude of the photon state. We have characterized the onset of the FSP with respect to the cavity frequency and the two-level energy spacing in the limit of small coupling strengths analytically. This analytical result for the regime that experiences population inversion agrees with the emergence of the FSP with an initial onset for resonant cavity frequency and Floquet energy difference.

The FSP presents a laser-like mechanism using population inverted Floquet states of two-level systems that are near resonance with a cavity mode. The model we proposed is in particular applicable to solid state systems coupled to a cavity, where the identical two-level systems are replaced by a momentum-dependent two-band model. The master equation

approach that we utilized is well-suited for describing such materials dissipatively. In such materials, Floquet state population inversion has been observed which provides motivation to implement this mechanism, with the prospect of creating Floquet-assisted laser systems.

## Acknowledgements

We thank Jayson Cosme and Jim Skulte for very helpful discussions.

**Funding information** This work is funded by the Deutsche Forschungsgemeinschaft (DFG, German Research Foundation) – SFB-925 – project 170620586, and the Cluster of Excellence 'Advanced Imaging of Matter' (EXC 2056), Project No. 390715994.

## A  Analytical Steady State Solutions

We take a two-level Hamiltonian $H = \vec{H}\vec{\sigma}$, such that $\text{Tr}(H) = 0$. Let $V$ be the transformation into the instantaneous eigenbasis of $H$, such that $VHV^\dagger = \epsilon\sigma_z$, where $\epsilon$ sets the energy scale of the Hamiltonian. In general such a Hamiltonian can be written as

$$H = \epsilon \begin{pmatrix} \cos(\theta) & e^{-i\phi}\sin(\theta) \\ e^{i\phi}\sin(\theta) & -\cos(\theta) \end{pmatrix} \tag{26}$$

such that

$$V = e^{i\sigma_y \frac{\theta}{2}} e^{i\sigma_z \frac{\phi}{2}}. \tag{27}$$

We write the Lindblad-von Neumann master equation in the original basis of $H$, but include dissipation in the instantaneous eigenbasis, such that $L_z = V^\dagger \sigma_z V = H\epsilon^{-1} = h$ and $L_\pm = V^\dagger \sigma_\pm V$. It is

$$\dot{\rho} = i[\rho, H] + \sum_{i \in \{+,-,z\}} \gamma_i (L_i \rho L_i^\dagger - \frac{1}{2}\{L_i^\dagger L_i, \rho\}) \tag{28}$$

$$= i\epsilon[\rho, h] + \gamma_z (\frac{1}{4}\text{Tr}(h\rho)h - \frac{1}{2}(\rho - \frac{\mathbb{1}}{2})) \tag{29}$$

$$+ \gamma_-(-\frac{1}{2}h - \frac{1}{2}(\rho - \frac{\mathbb{1}}{2}) - \frac{1}{4}\text{Tr}(\vec{h}\vec{\rho})h) + \gamma_+(+\frac{1}{2}h - \frac{1}{2}(\rho - \frac{\mathbb{1}}{2}) - \frac{1}{4}\text{Tr}(\vec{h}\vec{\rho})h) \tag{30}$$

with $\rho = \frac{1}{2}(\mathbb{1} + \vec{\rho}\vec{\sigma})$. We simplify this to

$$\partial_t(\vec{\rho}\vec{\sigma}) = i\epsilon[\vec{\rho}\vec{\sigma}, \vec{h}\vec{\sigma}] - \gamma_1 \vec{\rho}\vec{\sigma} - \gamma_2 \vec{h}\vec{\sigma} - \gamma_3(\vec{h}\vec{\rho})\vec{h}\vec{\sigma} \tag{31}$$

with

$$\gamma_1 = (\gamma_- + \gamma_+ + \gamma_z)/2 \qquad \gamma_2 = (\gamma_- - \gamma_+)/2 \qquad \gamma_3 = (\gamma_- + \gamma_+ - \gamma_z)/2 \tag{32}$$

and further

$$\dot{\vec{\rho}} = (2\epsilon(h \times \cdot) - \gamma_1 - \gamma_3 \vec{h}\langle\vec{h}, \cdot\rangle)\vec{\rho} - \gamma_2 \vec{h}. \tag{33}$$

We write $\vec{\rho}(t)$ with respect to the basis $\{\vec{h}, \dot{\vec{h}}, \vec{h} \times \dot{\vec{h}}\}$, such that

$$\vec{\rho}(t) = \rho_1(t)\vec{h} + \rho_2(t)\dot{\vec{h}} + \rho_3(t)(\vec{h} \times \dot{\vec{h}}) \tag{34}$$

$$\rho_1(t) = \vec{\rho}(t)\vec{h} \tag{35}$$

$$\rho_2(t) = |\dot{\vec{h}}|^{-2}\vec{\rho}(t)\dot{\vec{h}} \tag{36}$$

$$\rho_3(t) = |\dot{\vec{h}}|^{-2}\vec{\rho}(t)(\vec{h} \times \dot{\vec{h}}). \tag{37}$$

Assuming that $|\dot{\vec{h}}|^2$ does not depend on time, the equations of motion become

$$\dot{\rho}_1(t) = \partial_t(\vec{h}\vec{\rho}) = \dot{\vec{h}}\vec{\rho} + \vec{h}\dot{\vec{\rho}} = |\dot{\vec{h}}|^2\rho_2 - (\gamma_1 + \gamma_3)\rho_1 - \gamma_2 \tag{38}$$

$$\dot{\rho}_2(t) = |\dot{\vec{h}}|^{-2}\partial_t(\dot{\vec{h}}\vec{\rho}) = |\dot{\vec{h}}|^{-2}(\ddot{\vec{h}}\vec{\rho} + \dot{\vec{h}}\dot{\vec{\rho}}) = -2\epsilon(t)\rho_3 - \gamma_1\rho_2 + |\dot{\vec{h}}|^{-2}\ddot{\vec{h}}\vec{\rho} \tag{39}$$

$$\dot{\rho}_3(t) = |\dot{\vec{h}}|^{-2}\partial_t((\vec{h} \times \dot{\vec{h}})\vec{\rho}) = 2\epsilon(t)\rho_2 - \gamma_1\rho_3 + |\dot{\vec{h}}|^{-2}(\vec{h} \times \ddot{\vec{h}})\vec{\rho}. \tag{40}$$

We expand the second derivative of the Hamiltonian vector $\ddot{\vec{h}}$ in this basis as well and find

$$\ddot{\vec{h}}(t) = (\ddot{\vec{h}}\vec{h})\vec{h} + (\ddot{\vec{h}}\dot{\vec{h}})\dot{\vec{h}} + (\ddot{\vec{h}}(\vec{h} \times \dot{\vec{h}}))(\vec{h} \times \dot{\vec{h}}) \tag{41}$$

$$\ddot{\vec{h}}(t)\vec{\rho}(t) = \rho_1(\ddot{\vec{h}}\vec{h}) + \rho_2(\ddot{\vec{h}}\dot{\vec{h}})|\dot{\vec{h}}|^2 + \rho_3(\ddot{\vec{h}}(\vec{h} \times \dot{\vec{h}}))|\dot{\vec{h}}|^2 = -\rho_1|\dot{\vec{h}}|^2 + \rho_3(\vec{h}(\dot{\vec{h}} \times \ddot{\vec{h}})) \tag{42}$$

$$(\vec{h} \times \ddot{\vec{h}}(t))\vec{\rho}(t) = \rho_2((\vec{h} \times \ddot{\vec{h}}(t))\dot{\vec{h}})|\dot{\vec{h}}|^2 + \rho_3((\vec{h} \times \ddot{\vec{h}}(t))(\vec{h} \times \dot{\vec{h}}))|\dot{\vec{h}}|^2 = -\rho_2(\vec{h}(\dot{\vec{h}} \times \ddot{\vec{h}})). \tag{43}$$

We then arrive at the equations of motion

$$\dot{\rho}_1(t) = |\dot{\vec{h}}|^2\rho_2 - (\gamma_1 + \gamma_3)\rho_1 - \gamma_2 \tag{44}$$

$$\dot{\rho}_2(t) = -2\epsilon(t)\rho_3 - \gamma_1\rho_2 - \rho_1 + \rho_3|\dot{\vec{h}}|^{-2}\vec{h}(\dot{\vec{h}} \times \ddot{\vec{h}}) \tag{45}$$

$$\dot{\rho}_3(t) = 2\epsilon(t)\rho_2 - \gamma_1\rho_3 - \rho_2|\dot{\vec{h}}|^{-2}\vec{h}(\dot{\vec{h}} \times \ddot{\vec{h}}). \tag{46}$$

In the Rabi-problem in particular it is $\vec{H} = (\frac{E_\mathrm{d}}{\omega_\mathrm{d}}\cos(\omega_\mathrm{d}t), \frac{E_\mathrm{d}}{\omega_\mathrm{d}}\sin(\omega_\mathrm{d}t), \frac{\omega_z}{2})^\mathrm{T}$ and therefore

$$|\dot{\vec{h}}|^{-2}\vec{h}(\dot{\vec{h}} \times \ddot{\vec{h}}) = \frac{\omega_\mathrm{d}\omega_z}{2\sqrt{\frac{E_\mathrm{d}^2}{\omega_\mathrm{d}^2} + \frac{\omega_z^2}{4}}} \qquad |\dot{\vec{h}}|^2 = \frac{E_\mathrm{d}^2}{\frac{E_\mathrm{d}^2}{\omega_\mathrm{d}^2} + \frac{\omega_z^2}{4}} \qquad \epsilon(t) = \sqrt{\frac{E_\mathrm{d}^2}{\omega_\mathrm{d}^2} + \frac{\omega_z^2}{4}}, \tag{47}$$

which are all constant in time. We assume a periodic steady state $\rho(t) = \rho(t + \frac{2\pi}{\omega_\mathrm{d}})$ and express the equations of motion in terms of Fourier coefficients

$$im\omega\rho_1^m = |\dot{\vec{h}}|^2\rho_2^m - (\gamma_1 + \gamma_3)\rho_1^m - \gamma_2 \tag{48}$$

$$im\omega\rho_2^m = -2\epsilon\rho_3^m - \gamma_1\rho_2^m - \rho_1^m + \rho_3^m|\dot{\vec{h}}|^{-2}\vec{h}\vec{f} \tag{49}$$

$$im\omega\rho_3^m = 2\epsilon\rho_2^m - \gamma_1\rho_3^m - \rho_2^m|\dot{\vec{h}}|^{-2}\vec{h}\vec{f}. \tag{50}$$

We find that the Fourier modes do not couple in this representation. We solve the system of equations for arbitrary $m$ and find the complete expressions for $\rho_1^m$, $\rho_2^m$ and $\rho_3^m$, fully

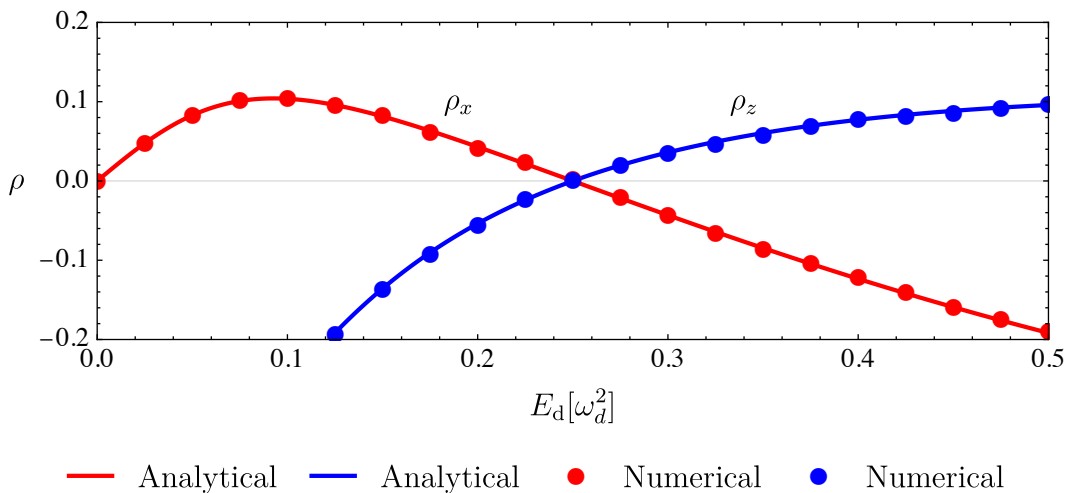

Figure 5: A comparison between the analytical (solid lines) and numerical (dots) results of the dissipative two-level steady state components $\rho_x$ and $\rho_z$ at $t = 2\pi\omega_\mathrm{d}^{-1}$ for $\lambda = 0$. It is $\omega_z = \omega_\mathrm{d}/2$. The zero-crossing of $\rho_z$ matches the onset of Floquet state population inversion in Fig. 3 (b).

determining the dissipative steady state $\vec{\rho}(t)$. For $m = 0$, expressed in the original basis, it is

$$\rho_x^0(t) = cE_\mathrm{d}\omega_\mathrm{d}^{-1}\left(\left(\gamma_1^2 + \omega_z^2 - \omega_\mathrm{d}\omega_z + 4E_\mathrm{d}^2\omega_\mathrm{d}^{-2}\right)\cos(\omega_\mathrm{d}t) + \gamma_1 E_\mathrm{d}\sin(\omega_\mathrm{d}t)\right) \tag{51}$$

$$\rho_y^0(t) = cE_\mathrm{d}\omega_\mathrm{d}^{-1}\left(\left(\gamma_1^2 + \omega_z^2 - \omega_\mathrm{d}\omega_z + 4E_\mathrm{d}^2\omega_\mathrm{d}^{-2}\right)\sin(\omega_\mathrm{d}t) - \gamma_1 E_\mathrm{d}\cos(\omega_\mathrm{d}t)\right) \tag{52}$$

$$\rho_z^0(t) = c\left(\left(\gamma_1^2 + (\omega_\mathrm{d} - \omega_z)^2\right)\omega_z - 2E_\mathrm{d}^2\omega_\mathrm{d}^{-2}(\omega_\mathrm{d} - \omega_z)\right), \tag{53}$$

with the prefactor

$$c = \frac{-\gamma_2\sqrt{\omega_z^2 + E_\mathrm{d}^2\omega_\mathrm{d}^{-2}}}{16E_\mathrm{d}^4\Gamma + \Gamma\omega_\mathrm{d}^4(\gamma_1^2 + (\omega_\mathrm{d} - \omega_z)^2)\omega_z^2 + 4E_\mathrm{d}^2\omega_\mathrm{d}^2(\gamma_1^2\Gamma + \gamma_1\omega_\mathrm{d}^2 + 2\Gamma\omega_z(-\omega_\mathrm{d} + \omega_z))} \tag{54}$$

and $\Gamma = \gamma_1 + \gamma_3$. In Fig. 5, we show the comparison between numerical results and the analytical solutions for $m = 0$, which match very well.

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
