# Peer review of "Floquet Engineering of Non-Equilibrium Superradiance"

_SciPost Physics_

## Round 1 · Referee Report · Anonymous (Referee 1) · 2022-4-19

Strengths

  • Timely and experimentally relevant
  • Floquet theory for driven-dissipative model

Weaknesses

  • Unclear if the described dynamical phase is new

Report

L. Broers and L. Mathey describe in their draft "Floquet Engineering of Non-Equilibrium Superradiance" the mean-field dynamics of a periodically driven and dissipative single-mode Dicke model. The authors show that this model exhibits beside the common superradiant phase of the dissipative Dicke model also a so-called Floquet-assisted superradiant phase (FSP). The latter is analyzed and predicted using Floquet theory. Moreover, the authors explain the macroscopic cavity field in the FSP by an occurring population inversion which is depleted due to superradiant photon emission.

The topic of their draft is timely and fits into the very active research field of superradiance. The analysis appears to be correct and rigorous up to some minor problems (see below). However, the main problem of this manuscript is how the here observed FSP compares to the various dynamical superradiant phases that have been studied recently. For example, is the FSP fundamentally different from the D-NP phase that is studied in PRA 92, 023815 (2015)? In general, a clear comparison to existing literature seems to be missing and is needed to justify why this work is of interest for the physics community. Without such a justification I cannot support publication in SciPost Physics.

Beside this general criticism I have some further comments and questions below:

(1) One motivation of this work seems to be the "Floquet engineered light-amplification in solids" and it is argued that the two-level description, which is used in the manuscript, can be applied to two bands models such as monolayer graphen. I would be interested in what the underlying approximation is? Do you require that the band gap is much larger than the bandwidth or something similar? Could you make a remark on (your expectations) how inhomogeneous broadening (e.g. due to motion) would affect the FSP phase and the coherence of the light?

(2) Your mean-field description includes various kinds of dissipation. I did not really understand why you use "dissipation of the two-level system in its instantaneous eigenbasis" and not in a stationary basis. Is there a physical reason? Does the FSP require such a model and do you require dissipation at all for the FSP?

(3) In Appendix A, you provide the derivation of the master equation for rho. When I tried to derive the term proportional to gammaz in Eq. (29) I got an additional factor of 4. Maybe you could check this again, but I think it would anyway only rescale gammaz.

(4) It would be good to explicitly write in the Appendix why you are doing the calculations below Eq. (33). From my understanding this is needed for the derivation of the lambda=0 case (that you require in Fig.3 and Eq.(22-25)) but this is not really written in the Appendix and only briefly referenced before Eq.(23).

(5) Before Sec. 3 you write: "The cavity loss rate is small compared to what is commonly referred to as the good cavity regime". There seems to be a frequency missing? Is the "good cavity regime" required? How should the linewidth compare to the bandwidth of a solid-state system?

(6) Could you show Fig. 2(c) within a smaller parameter region of omega (y-axis)? At the moment most of this subplot is white. Is the width in frequency omega only limited by the integration time of Eqs. (4) and (5)? Do you expect that the linewidth of the emitted light in the FSP can overcome the linewidth of the driving field?

(7) You make the statement before Eq. (21) that "slight mismatches are a consequence of the approximation made to justify the expression of HF". Are you here referring to the assumption that alpha oscillates with omegac. In that context, you compare the Floquet energies with the maxima of n(omega) and they agree quite well. Do these Floquet energies also explain the curvature of the maximum of alpha(omega) in Fig. 2(c) which is not exactly at omegac?

(8) In Eq. (18) for H1 you use the absolute value |alpha|. How do you calculate |alpha| here? Do you solve for alpha numerically or do you solve Eq. (4) using dalpha/dt=-i omegac alpha? Is the complex argument of alpha arbitrary? If yes, what breaks this symmetry in your simulation/model?
  • validity: high
  • significance: good
  • originality: good
  • clarity: high
  • formatting: good
  • grammar: good

Author:  Lukas Broers  on 2022-08-30  [id 2771]

(in reply to Report 1 on 2022-04-19)

We have attached our reply to this comment.

Attachment:

referee_1_reply.pdf

---

## Round 1 · Referee Report · Anonymous (Referee 2) · 2022-5-24

Report

The manuscript by Broers and Mathey discusses the emergence of
superradiant phases in a off-resonantly driven Dicke model. The
realization of such non-equilibrium states of matter is a very active
field, and the fact that the authors relate their results to the
well-known and widely studied Dicke model makes the manuscript
potentially of interest for SciPost Physics. However, there is a
potential issue that the authors need to address before I can actually
recommend publication:

The actual parameters required for the realization of the Floquet
superradiant phase are quite unclear. First, the units used in
Eq. (1) are not obvious. From Fig. 2, it appears that the model is
operated in a strong driving regime, as E_d is given in units of
omega_d^2. Additionally, the coupling strength lambda is always
given in units of its critical value, which makes it impossible to
relate it to actual physical quantities.

My main concern is that in the parameters required to observe Floquet
superradiance, the usual assumptions for the derivation of a Lindblad
form (weak coupling, separation of timescales) might no longer
hold. To some extent, the authors seem to aim to address this by
working in the instantaneous eigenbasis, but it is not clear how this
has been derived and in which limits Eq. (5) remains valid.

Other than that, the manuscript is well written and the results are
convincing.
  • validity: -
  • significance: -
  • originality: -
  • clarity: -
  • formatting: -
  • grammar: -

Author:  Lukas Broers  on 2022-08-30  [id 2772]

(in reply to Report 2 on 2022-05-24)

We have attached our reply to this comment.

Attachment:

referee_2_reply.pdf

---

## Editorial Decision

resubmitted